# Enhanced Succinate Oxidation with Mitochondrial Complex II Reactive Oxygen Species Generation in Human Prostate Cancer

**DOI:** 10.3390/ijms232012168

**Published:** 2022-10-12

**Authors:** Aijun Zhang, Anisha A. Gupte, Somik Chatterjee, Shumin Li, Alberto G. Ayala, Brian J. Miles, Dale J. Hamilton

**Affiliations:** 1Center for Bioenergetics, Houston Methodist Research Institute, Houston, TX 77030, USA; 2Department of Medicine, Houston Methodist, Weill Cornell Medicine Affiliate, Houston, TX 77030, USA; 3Department of Pathology and Genomics, Houston Methodist Hospital, Weill Cornell Medicine Affiliate, Houston, TX 77030, USA; 4Department of Urology, Houston Methodist Hospital, Weill Cornell Medicine Affiliate, Houston, TX 77030, USA

**Keywords:** citrate, TCA cycle, electron transport system, respiratory analysis

## Abstract

The transformation of prostatic epithelial cells to prostate cancer (PCa) has been characterized as a transition from citrate secretion to citrate oxidation, from which one would anticipate enhanced mitochondrial complex I (CI) respiratory flux. Molecular mechanisms for this transformation are attributed to declining mitochondrial zinc concentrations. The unique metabolic properties of PCa cells have become a hot research area. Several publications have provided indirect evidence based on investigations using pre-clinical models, established cell lines, and fixed or frozen tissue bank samples. However, confirmatory respiratory analysis on fresh human tissue has been hampered by multiple difficulties. Thus, few mitochondrial respiratory assessments of freshly procured human PCa tissue have been published on this question. Our objective is to document relative mitochondrial CI and complex II (CII) convergent electron flow to the Q-junction and to identify electron transport system (ETS) alterations in fresh PCa tissue. The results document a CII succinate: quinone oxidoreductase (SQR) dominant succinate oxidative flux model in the fresh non-malignant prostate tissue, which is enhanced in malignant tissue. CI NADH: ubiquinone oxidoreductase activity is impaired rather than predominant in high-grade malignant fresh prostate tissue. Given these novel findings, succinate and CII are promising targets for treating and preventing PCa.

## 1. Introduction

PCa is the most common malignancy and the second leading cause of cancer death in men in the United States [1]. The incidence of aggressive metastatic disease in the 55–69-year age group has increased since 2007 [2]. Conservative strategies with watchful waiting and active surveillance have emerged as the primary management for the majority of low-grade clinical staged PCa [3]. However, follow-up biopsies in more than 12% of those undergoing active surveillance document Gleason Score (GS) progression to a higher risk grade [4]. Although there have been attempts to identify the mechanism and markers for progression, the tumor metabolism in this subset of high-risk patients is poorly understood. Metabolically, a transformation from citrate-producer to citrate-oxidizer has been described as the metabolic phenotype that differentiates transformed PCa [5,6,7]. The metabolic phenotype of prostate is unique: the high concentration of zinc inhibits mitochondrial m-aconitase (EC 4.2.1.3), which initiates the oxidation of citrate in the first span of the tricarboxylic acid cycle (TCA) in non-cancerous prostate tissue [8,9]. This unique metabolism carries out the primary physiological function of citrate secretion [10,11]. With malignant transformation, the decline in zinc concentration reduces the inhibition of m-aconitase and increases citrate oxidative flux. With unrestrained citrate oxidation, one would anticipate an increase in NADH-mediated mitochondrial complex I (CI) electron transport activity [10]. Most evidence is based on gene expression [12,13], and mtDNA assessments have been areas of focus [14,15]. There is evidence from tissue data sets [16] and circulating mtDNA analysis [17], including changes in peripheral leukocyte mtDNA copy number [18], that mitochondrial alterations are associated with tumor aggressiveness and risk category. This includes mitochondrial fusion and electron transport system (ETS) alterations involving NADH: ubiquinone oxidoreductase (EC 1.6.5.3) CI [19]. In established cell lines, there is evidence for increased succinate with increased succinate-supported respiration [20]. A publication assessing relative oxidation of fatty acids in cell lines and specimens of benign prostate and PCa from six glands documented increased beta-oxidation of fatty acids in the former. That study noted the effect of succinate to be higher [21]. Human tissue studies using frozen stored specimens have documented increased reactive oxygen species (ROS) levels [22,23]. These and other studies provide indirect evidence for PCa tumorigenesis associated with mitochondrial content and functional changes [24], but they lack direct evidence in fresh human PCa tissue.

In this study, we aim to demonstrate that PCa is associated with mitochondrial energy transfer inefficiencies that impact substrate oxidation capacity and induce redox alterations that promote neoplastic growth and aggressiveness. We present comparative respiratory findings of paired mechanically permeabilized fresh non-malignant and malignant prostate tissues. Our findings document dominant mitochondrial succinate SQR (EC 1.3.5.1) CII oxidation in PCa with evidence of enhanced CII ROS production and impaired CI oxidation in advanced GS–grade tissue.

## 2. Results

### 2.1. Patient Demographics and Biopsy Pathology

Five prostate peripheral zone punch biopsies [25,26] were procured from each of 23 PCa patients immediately following prostatectomy. Each biopsy was split into three sections: one for high-resolution respirometry (HRR) study, one for mRNA expression, and a third for histopathological diagnostics (Figure 1A). All the experimental results were analyzed and correspond with the pathologic diagnosis of each biopsy (Figure 1B).

Appendix A presents patients’ demographics and prostate sample characteristics. Twenty-seven subjects aged 53 to 76 years with clinical pre-operative biopsy GS’s ranging from 3 + 3 to 4 + 5 were consented and enrolled in the study. One gland could not be processed in a timely fashion following resection, and surgery was postponed on three other subjects. Therefore, tissue biopsies were procured from 23 of the 27 enrollees. Individual GS grades were determined by independent blinded analysis of hematoxylin and eosin stained (H&E) slides of each biopsy specimen (Figure 1B).

### 2.2. The Oxidative Phosphorylation (OXPHOS) Rate of CII (P_CII_) Is Higher Than That of CI (P_CI_) in Non-Malignant Tissue

OXPHOS was analyzed with HRR by sequentially assessing CI, CI + CII and CII pathways. The oxygraphy recordings from CI + CII and CII + rotenone protocols are shown in Figure 2A,C, and the pathways in Figure 2B,D, respectively. Relative CI and CII convergent electron flow into the Q-junction was assessed by comparing the OXPHOS (P) ADP-activated oxygen consumption rate, the uncoupled ETS (E) respiratory capacity rate and the relative LEAK (L) oxygen flux without ADP activation. 

The non-malignant tissue CI path P rates (P_CI_, Figure 2A) were lower than the CII P (P_CII_, Figure 2C) in non-malignant specimens sourced from both lower GS 3 + X and higher-grade GS 4 + X PCa–containing glands (Table 1). The non-malignant P_CII_ comprised > 60% of the P_CI+CII_ rates. In other words, P_CII_ contributed much more than P_CI_ to the P_CI+CII_, which would be consistent with zinc inhibiting citrate oxidation activity. The inhibition of m-aconitase (EC 4.2.1.3) by the high concentration of zinc in prostate epithelial tissue would reduce NADH delivery to CI, consequently redirecting pyruvate for citrate synthesis and excretion.

Furthermore, we observed the P/E control ratio, which indicates the limitation of OXPHOS capacity by the phosphorylation system. In specimens from low-grade (GS 3 + X) glands, the E_CII_ uncoupled respiratory rate of 11.98 ± 1.15 pmol/(s·mg) tissue was not significantly higher than the P_CII_ rate of 11.26 ± 1.31 pmol/(s·mg) (Table 1, Figure 2C), so the P/E = 0.94. This means that the CII OXPHOS capacity was not limited by the mitochondrial phosphorylation system. However, the E_CI+CII_ rate 19.07 ± 1.40 increased from P_CI+CII_ 16.54 ± 1.31 pmol/(s·mg) (Table 1, Figure 2A), so the CI + CII P/E ratio declined to 0.86. This indicates that the CI OXPHOS (it should be CI + CII OXPHOS here, but we say CI because the CII was not limited) capacity was limited by the phosphorylation system, compared with the CII P/E ratio of 0.94 for non-malignant tissue. This is consistent with the inhibited CI by high-concentration zinc observed in prostate tissue [27].

### 2.3. PCa GS Grade Enhances CII Flux with Increased L/P Coupling Ratio

Comparative paired non-malignant/malignant tissue assessment in 9 GS 3 + X and 6 GS 4 + X PCa prostate glands demonstrated CII succinate oxidation as the dominant electron source for coupled phosphorylation P and uncoupled electron transfer E. The P_CI_ did not differ much in each group, but the P_CI + CII_ increased more in the malignant group than in the non-malignant. In GS 4 + X especially, the P_CI + CII_ malignant (21.32 ± 2.25 pmol/(s·mg)) was significantly greater than the non-malignant (15.67 ± 1.61 pmol/(s·mg)) (Table 1). There was a GS-related increase in the GS 4 + X malignant vs. non-malignant CI + CII:CI ratio (P_CI + CII_:P_CI_) in the paired tissue analysis (Figure 3A). The GS 3 + X and 4 + X P_CI + CII_:P_CI_ average ratios were 4.05 and 4.00, respectively, in the malignant groups, and the respective non-malignant ratios were 3.01 and 3.1 (Table 1, Figure 3A). P_CII_ was 13.48 ± 2.85 pmol/(s·mg) in GS 4 + X malignant, which increased than in other groups. These findings reflect the enhanced P_CII_ in malignant tissue. These findings are consistent with those in non-malignant tissue reported by Schöpf et al. [27].

The LEAK, an indirect measure of potential ROS formation, in CI (L_CI_) was less than L_CII_ in the presence of succinate and rotenone. The L_CII_ in the GS 4 + X malignant group (8.70 ± 2.58) was significantly greater than in the non-malignant group (5.27 ± 0.85) (Table 1). The L/P coupling controls the LEAK to OXPHOS ratio and is the inverse of the respiratory control ratio (RCR). It reflects the coupling efficiency and the limitation by the phosphorylation system. The average CII L/P ratio rose from GS 3 + X 0.47 to GS 4 + X 0.65, and paired CII L/P analysis confirmed a significant increase (Figure 3B). The malignant E_CII_ rates were also higher than in non-malignant tissue. The evidence points to increased CII substrate oxidation with enhanced CII L in cancer tissue, which suggests CII-sourced ROS production.

### 2.4. TCA Metabolomics Support Respiratory Finding of Succinate CII Dominant Flux

Comparative analysis of relative tissue concentrations of metabolic intermediates pyruvate, citrate, cis-aconitate, alpha-ketoglutarate, succinate, fumarate, malate, oxaloacetate, and lactate in GS 3 + X and GS 4 + X malignant vs. non-malignant paired same-gland specimens was performed. As shown in Figure 4, the citrate and cis-aconitate were higher in GS 3 + X malignant than in non-malignant tissue, indicating higher CI capacity, which is the electron entry to the Q junction through CI [28,29]. The citrate and cis-aconitate were significantly lower in the GS 4 + X malignant than in GS 3 + X malignant specimens, but the succinate was significantly higher and the P_CII_ enhanced. We can see that succinate and fumarate are higher in GS 4 + X malignant tissue. These data are consistent with the above respiratory findings of dominant succinate oxidative flux through the later span of the TCA. 

### 2.5. GS Grade Is Related to CII Forward ROS Production Rate

With the finding of an increased CII LEAK (L_CII_) rate, we performed 30 min Amplex UltraRed tissue assays for H_2_O_2_ concentration (µmol/mg tissue) comparing non-malignant and malignant specimens from 11 glands with pre-operative GS grades 3 + X and 4 + X. The malignant specimens contained a significantly greater concentration of H_2_O_2_ than the non-malignant specimens in each group (Figure 5A). The GS 4 + X non-malignant group had a significantly greater H_2_O_2_ concentration than the GS 3 + X non-malignant group, as well. Immunohistochemical analysis with anti-4-hydroxynonenal (4-HNE) antibody assay on GS 4 + X compared with a negative control (Figure 5B) provided evidence of lipid peroxidation in the higher-grade tumor specimens consistent with an increased redox state.

## 3. Discussion

The procurement and timely processing of fresh human PCa tissue for respiratory analysis presents challenges. There are very few published studies assessing fresh tissue mitochondrial respiratory function, especially comparing paired non-malignant and malignant tissue. The time from procurement in the surgical suite to transport and preparation for laboratory respiratory analysis is a key limiting factor for fresh tissue respiratory study. These challenges require the coordinated effort of a multi-disciplinary team. With PCa, cancerous tissue is not often apparent by gross examination without fixative. Therefore, for this study, we obtained and labeled four to five left to right punch biopsies from the peripheral zone, which is frequently seeded with cancer foci [30]. The methods for respiratory analysis included mechanical permeabilization and protocols as published by Schöpf et al. [27]. Timely accurate processing required coordinated collaboration with surgical and pathological specialists beginning in the operating room suite. Later, each biopsy slide was GS graded by a prostate pathology specialist for correlation with the mitochondrial respiratory findings. The paired non-malignant control specimens cannot be considered normal prostate tissue, as they were biopsied from a gland with multiple foci of PCa.

Normal prostate epithelial tissue has a unique TCA cycle flux profile. Zinc is concentrated in the mitochondrial matrix, where it inhibits the activity of m-aconitase (EC 4.2.1.3), the enzyme that catalyzes citrate isomerization to cis-aconitate and isocitrate [8]. The formation of isocitrate is non-oxidative, but it undergoes oxidative decarboxylation by isocitrate dehydrogenase (EC 1.1.1.42) with the reduction of NAD+ to NADH. Hence, citrate oxidized in the first span of the TCA cycle results in NADH production, which would then be oxidized and increase electron flux through CI into the Q-junction and entry into the ETS. A transition from citrate secretion to TCA oxidation would thereby increase CI oxidative flux (CI P) as a result of reduced zinc concentration. These metabolic signatures of the prostate and cancer transformation have attracted the attention of scientists over the past few years. Complex I has been a focus for PCa therapy [31,32], but very few studies of fresh prostate tissue have been done because of the aforementioned challenges. The current study conducted respiratory analysis using gated and convergent multiple substrate protocols on paired same-gland non-malignant and PCa fresh tissue specimens, but the anticipated enhanced P_CI_ was not observed in the PCa samples. Rather, we observed enhanced P_CII_ resulting from succinate oxidation. The respiratory results of non-cancerous specimens are consistent with published results on benign human tissue by Schöpf et al. [27]. Associated with increased CII flux, we found evidence of GS-related enhanced CII-sourced ROS production (Figure 6). Results published by Schöpf et al. [29] in PCa tissue document mutations in mitochondrial CI that are associated with a reduction in NADH oxidation pathway, which in turned could trigger a transition to increased CII oxidative flux. The metabolic studies document relative reduction of citrate and aconitate with relative increase succinate-fumarate-oxaloacetate in the GS 4 + X graded malignant tissue (Figure 4). In these mechanically permeabilized samples, the complex structure limits metabolite localization to the tissue rather mitochondrial level. Other source(s) of succinate are under consideration for ongoing investigations. Comparing these fresh tissue studies with the zinc—CI studies, which were mostly completed on human prostate cancer cell lines (PWPE-1, PC-3, LNCaP, etc.), frozen human prostate tissue, and animal models—the metabolism is different, especially related to P_CI_. Variables in cell lines and animal models are easier to control in the experiments, but cell lines and animal models cannot represent the interdependence and interactions with other cells, stroma, and organs. The fresh tissue provides insights into PCa in relation to its microenvironment.

Directly measuring H_2_O_2_ using an Amplex UltraRed protocol documented higher H_2_O_2_ production in GS 4 + X than in GS 3 + X malignant specimens. This was supported by the immunohistochemistry results with anti 4-HNE staining and by analysis of relative respiratory L/P ratios. The L/P coupling ratios with CI substrates glutamate + malate did not differ in paired specimens. However, the paired non-malignant and PCa tissue analysis with the CII substrate protocol of succinate + rotenone documented a significant difference between the higher-grade GS 4 + X and the non-malignant and lower-grade GS 3 + X specimens. The increased LEAK oxygen consumption rate as measured in the presence of substrate with the absence of ADP correlates with ROS production, which would result from electron leak. Other factors could include uncoupled proton leak, electron and proton slippage which would reduce protonmotive force (ΔP) [33,34]. The current findings are supportive of a CII contribution to the PCa oxidative state likely through electron leak with superoxide formation. Documented CII sites of ROS production include the flavin component, the ubiquinone binding site and possibly the [3Fe-4S] iron-sulfur clusters in both the forward and reverse CII directions [35,36]. In our PCa specimens, there is evidence of CII ROS production in the forward direction, especially in higher GS grade specimens. The concentration of succinate would limit reverse flow, supporting ROS production in the CII forward direction. Although reverse electron flow into complex I has been documented [37,38] and discounted [39], the addition of rotenone in this study would inhibit a reverse CI ROS source. Further studies are necessary for confirmation and site localization.

The metabolomics data show the potential for enhanced CI flux in low-grade GS 3 + X malignant tissue. Low-grade and early state PCa cells are in close contact with the normal vascular nutrient supply. The zinc concentration declines in this PCa, and the cells have the potential for an increase in energy production through CI. The elevated tissue citrate concentration as shown in Figure 4 can inhibit PFK1 which in turn would suppress glycolysis and the Warburg effect and contribute to reduced FDG-PET sensitivity in low grade PCa [40,41,42]. The higher-grade or later stage of PCa cells, in contrast, outgrow that supply and remodel their metabolism by enhancing CII to support the PCa’s aggressive proliferation [28,43]. The relative reduction in GS 4 + X citrate concentration shown in Figure 4 is consistent with published findings of enhanced citrate lyase conversion of cytosolic citrate to acetyl-CoA for lipid synthesis in cancers [44,45].

This is one of very few studies designed to compare mitochondrial respiratory function in paired non-malignant and PCa specimens from freshly procured human prostate. The design was based on an earlier published comparative cell line and fresh human prostate tissue study [21]. That investigation documented a metabolic distinction related to fatty acid metabolism and respiratory function between established cell lines and fresh human prostate tissue. 

There are a number of key findings in this current investigation: first, the unanticipated enhanced mitochondrial CII (SQR), rather than CI flux, in the fresh prostate cancer tissue. Malignant transformation to PCa has been metabolically characterized as an enhanced P_CII_ and impaired P_CI_. Furthermore, in support of the increased CII flux, our findings include documenting enhanced CII ROS production in higher-grade PCa. This supports our hypothesis that mitochondrial energy transfer inefficiencies with associated substrate oxidation adaptations play a role in PCa aggressiveness. Further study is required. This is one of only a few published studies to assess comparative respiratory flux in fresh human prostate malignant and non-malignant tissue.

An understanding of stage-based mitochondrial energy transfer mechanisms in PCa would offer insights for patient-centered diagnostic development and provide background for targeted metabolic therapy aimed at mitochondrial substrate selection and ROS production. 

## 4. Materials and Methods

### 4.1. Subjects

The study protocol for tissue and blood specimen collection and the experimental methods were approved by the local Houston Methodist ethics committee and Institutional Review Board ID—Pro00017176:1. The research protocol, methodologies and consenting process complied with the code of ethics and standards of the Declaration of Helsinki. Patients undergoing radical prostatectomy by a urological surgical specialist (BM) were screened and recruited pre-operatively. Screening criteria included biopsy evidence of treatment-naïve prostate adenocarcinoma, GS 3 + 3 and higher. Each subject was provided with a study protocol description and interviewed to explain and answer questions. Each individual prospectively provided written informed consent for participation and use of venous blood samples and resected tissues. Collected information included medical and clinical data that were de-identified and assigned a study number. Sampling included pre-operative venous blood sampling and post-operative prostate tissue collection following radical prostatectomy. The specimens were promptly de-identified and assigned a study number.

### 4.2. Tissue Procurement and Processing

The laparoscopic robotic assisted radical prostatectomy procedure was completed by an experienced surgeon (BM). Immediately following resection of the prostate with intact seminal vesicles, the specimen was placed in a container with ice in the surgical suite then promptly transported to an adjacent processing room. Under the supervision of pathology staff, the peripheral zone was chosen for biopsy, because it is the site most often harboring malignancy [25,26]. Four to five 4 mm diameter and 8 mm length punch biopsy specimens were chosen under direction of the urologic pathology specialist. Each biopsy is approximately 90–110 mg wet weight (Figure 1). These biopsies were de-identified, then transported on ice within 10 min to the research laboratory. Each punch biopsy specimen was then sliced into three sections: one was for respiratory study, one was snap-frozen for later gene and protein analysis, and a third was fixed in 10% formalin and embedded in paraffin for future histological examination. The respiratory study specimens with wet weights of approximately 40 mg were promptly mechanically permeabilized. Samples were equally distributed for separate HRR protocols as described below and reported in Schöpf et al. [27].

### 4.3. Histopathology and Immunohistochemistry

Each of the four to five punch biopsy specimens was processed and H&E stained using standard procedures by our pathology core facility. The immunohistochemistry staining and imaging were completed in our laboratory. In brief, after microdissection, the tissue specimens were fixed in 10% formalin for 24 h. They were processed by a histological processing instrument, then embedded in paraffin. Sections 5 µm thick were stained with H&E. Slide interpretation, malignancy and corresponding GS grade in each prepared specimen was independently determined by a single prostate pathology specialist (AA) blinded to the corresponding metabolic and bioenergetic findings (Figure 2). Likewise, the pathologic interpretations of the specimens were unknown to the research team performing the respiratory and ROS studies.

Immunohistochemistry and immunofluorescence were performed with anti-4-HNE (Abcam, Waltham, MA, USA) on the same tissue specimen we used for H&E histological evaluation. The staining kits and reagents were Dako system (Agilent Technologies, Inc., Santa Clara, CA, USA). The stained slides were scanned using the Nikon Eclipse Ti fluorescence microscope (Nikon, Tokyo, Japan) with a 20× objective. The image was acquired and processed with NIS-Elements AR 4.00.07 software (Nikon, Tokyo, Japan).

### 4.4. Respirometry Protocols

Each of the four to five respiratory specimens was placed in iced relaxing and preservation BIOPS solution (2.8 mM CaK_2_egtazic acid [EGTA], 7.2 mM K_2_EGTA, 5.7 mM Na_2_ATP, 6.6 mM MgCl_2_·6H_2_O, 20 mM taurine, 15 mM Na_2_phosphocreatine, 20 mM imidazole, 0.5 mM dithiothreitol, 50 mM 2-[N-morpholino] ethanesulfonic acid [MES], pH 7.1), then mechanically permeabilized and dissected into 9–12 mg wet-weight specimens while in BIOPS solution for the protocoled HRR study [46]. Oxygen consumption studies were completed using Oxygraph-2K (Innsbruck, Austria) for HRR quantification under a constant temperature of 37 °C and stirring at 750 rpm. Real-time data were fed to DATLAB 6 software (Oroboros Instruments, Innsbruck, Austria) for post-experimental analysis. Oxygen concentration was maintained between 200 and 300 µM. Oxygen consumption was calculated as pmol/(s·mg) wet tissue mass. Sequential manual titrations using Hamilton microsyringes of the substrate concentrations for the two substrates-uncoupler-inhibitors protocols listed in Table 2 were applied to each mechanically permeabilized biopsy specimen. Cytochrome c (10 µM) was pipetted during the ADP (2.5 µM) activated state of each protocol to assess outer mitochondrial membrane integrity.

The first protocol sequence was designed to assess CI and/or CII with multi-substrate, inhibitor, and uncoupler titrations of glutamate-malate (GM), ADP, cytochrome c, pyruvate, succinate, FCCP, rotenone, malonate and antimycin A (Table 2). The second protocol, designed to assess complex II substrate flux control, included sequential addition of rotenone, succinate, ADP, cytochrome c, FCCP, malonate and antimycin A (Table 2). The substrate, uncoupler and inhibitor concentrations are documented in Table 2. LEAK respiration during state 2 (L), coupled respiration state 3 (P) and uncoupled rates E were documented. The first protocol included CI and CII assessments with LEAK respiration after GM and P, then OXPHOS respiration after ADP. CII + CI convergent flow was assessed after the addition of pyruvate and succinate. FCCP was then titrated to assess phosphorylation limitation, followed by CI inhibitor rotenone to distinguish the relative contribution of CII to respiratory or ETS capacity. Sequential inhibitors of CII, malonate, and complex II antimycin were added for ETS blockade to assess residual oxygen consumption. The second protocol included CII assessment alone.

### 4.5. ROS Analysis

The hydrogen peroxide (H_2_O_2_) level in prostate tissue was quantified using the Amplex UltraRed Hydrogen Peroxide/Peroxidase Assay Kit (ThermoFisher Scientific Inc., Waltham, MA, USA) per the instructions of the manufacturer. Briefly, the tissue specimens were mechanically and chemically (saponin, 50 µg/mL for 30 min) permeabilized in ice-cold reaction buffer. Then, 5 mg/well of each specimen were placed into a 96-well plate in duplicates [46]. The specimens were incubated in the working solution (100 µM Amplex UltraRed, 0.2 U/mL horseradish peroxidase, 1000 U/mL superoxide dismutase) at room temperature for 30 min. The fluorescence was measured with Synergy HT microplate reader (BioTek, Winooski, VT, USA) at excitation and emission wavelengths of 563 and 587 nm. The results were correlated with the cancer GS from the prostate gland pathological analysis.

### 4.6. Targeted Metabolomics

Following harvest, 20 prostate tissue specimens were stored at −140 °C in liquid nitrogen until analyzed. Targeted metabolomic assays of non-malignant and prostate cancer tissue homogenates were performed at the Metabolomics Center, Baylor College of Medicine. The dried extract was resuspended in injection solvents and analyzed using ultrahigh performance liquid chromatography (UPLC) coupled to tandem mass spectrometry (Agilent 1290 series UPLC system). The assays included measurement and normalization of TCA intermediates including citrate, cis-aconitate, alpha-ketoglutarate, succinate, fumarate, malate, and oxaloacetate. Pyruvate and lactate metabolites were also assayed. A total of 19 metabolites from two methods were assayed. The first, Water-Neg, was normalized by Internal Standard L-Tryptophan (ISTD), and the second, TCS Glycolysis, was normalized by Internal Standard L-Gibberilic acid (ISTD). Cancer and non-cancer specimens were compared based on Gleason score.

### 4.7. Statistical Analysis

Data from all the experiments were calculated as means ± standard deviation. Comparisons between two groups were computed with two-tailed unpaired *t*-tests. For paired specimens from the same gland, the paired *t*-test and the ANOVA with repeated measurement were used for comparisons between more than two groups. All tests were completed using GraphPad Prism 8.4.3 for Windows statistical software (GraphPad Software, San Diego, CA, USA). Significance was accepted for *p* < 0.05. 

## Figures and Tables

**Figure 1 ijms-23-12168-f001:**
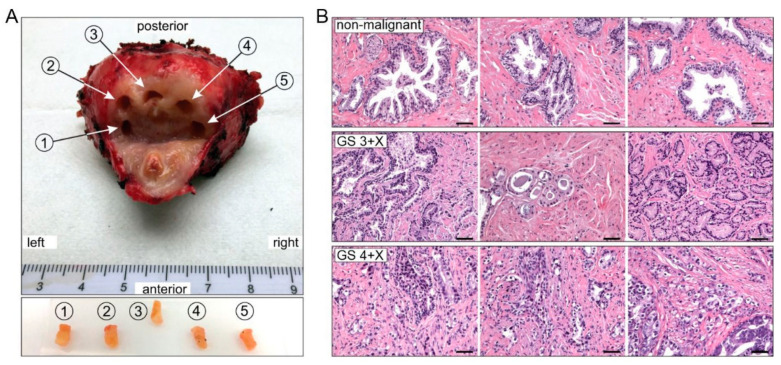
Prostate biopsy and tissue histopathology confirmation. (**A**) Punch biopsies were procured from the prostate peripheral zone and numbered 1–5, from left to right. (**B**) Histopathology representative images (hematoxylin and eosin staining, H&E). Upper row is non-malignant histology; middle row is GS 3 + X and lower row is GS 4 + X malignant. Scale bar = 100 µm.

**Figure 2 ijms-23-12168-f002:**
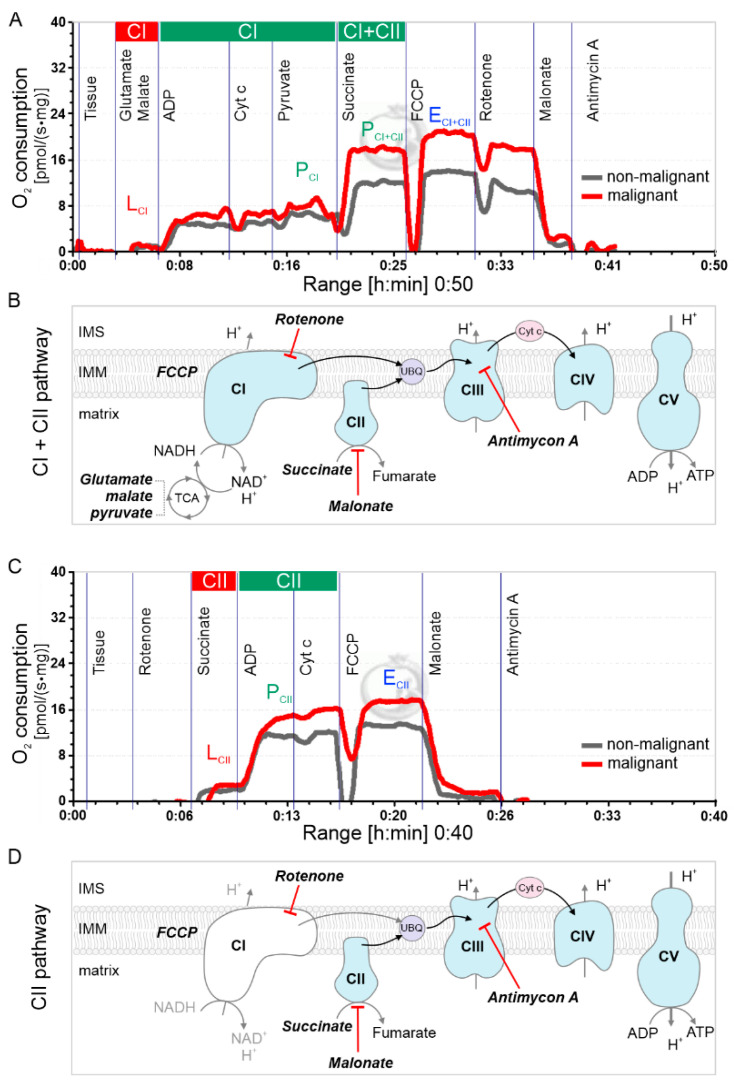
High-resolution Oxygraph recording examples from prostate non-malignant and malignant mechanically permeabilized tissue specimens. Blue line indicates time-dependent oxygen concentration nmol/mL (left Y axis,) and red line indicates oxygen consumption per second and per mg wet weight (right Y axis). (**A**) Protocol CI + CII to assess relative convergence of CI and CII to ubiquinone Q-junction. Sequential substrate additions (see Table 2) indicated. Grey line indicates oxygen consumption per second and per wet weight of non-malignant tissue and red line indicates malignant tissue. (**B**) CI + CII pathway, the substrates and inhibitors highlighted in bold and italic. IMS: intermembrane space, IMM: inner mitochondrial membrane, matrix: mitochondrial matrix. (**C**) Protocol CII + rotenone for assessment of CII-linked respiration in non-malignant and malignant tissue. Cytochrome c was used to assess intactness of the outer mitochondrial membrane. (**D**) CII pathway, the substrates and inhibitors highlighted with board font. Cytochrome c was used to assess intactness of the outer mitochondrial membrane.

**Figure 3 ijms-23-12168-f003:**
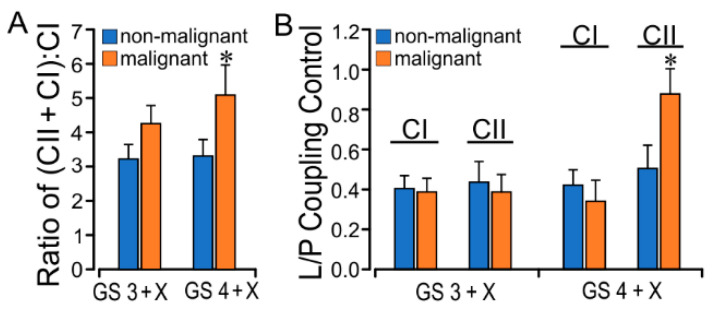
Comparative non-malignant/malignant tissue OXPHOS and LEAK to OXPHOS (L/P) ratios. (**A**) Paired OXPHOS ratios of (CII + CI):CI (P_CI + CII_:P_CI_)for non-malignant/malignant vs. GS 3 + X/GS 4 + X. (**B**) CI and CII L/P coupling control ratios for non-malignant/malignant vs. GS 3 + X and GS 4 + X prostate cancer. Significantly increased higher-grade (GS 4 + X) malignant tissue ratios indicate enhanced CII relative to CI OXPHOS associated with increased CII LEAK. * *p* < 0.05.

**Figure 4 ijms-23-12168-f004:**
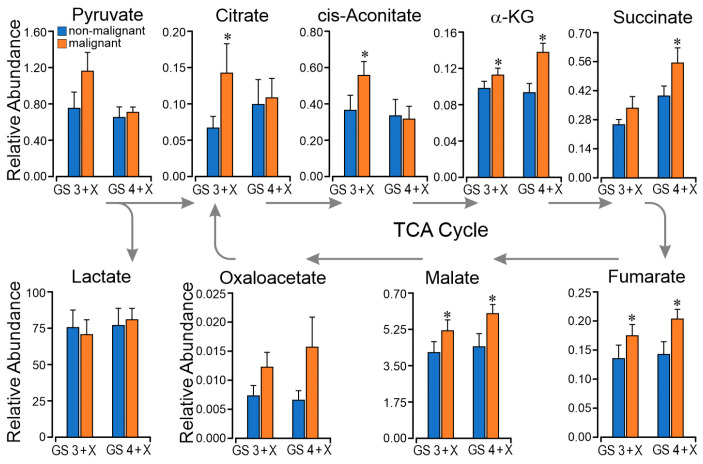
Metabolomics study comparing paired non-malignant/malignant in GS 3 + X and GS 4 + X graded specimens. The metabolite results aligned as in the TCA cycle. * *p* < 0.05.

**Figure 5 ijms-23-12168-f005:**
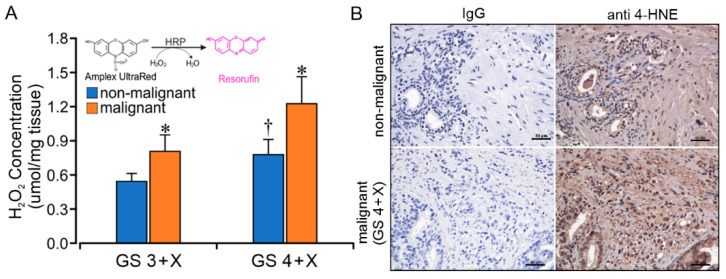
The reactive oxygen species (ROS) production and lipid peroxidation in prostate tissue. (**A**) GS 3 + X and GS 4 + X graded tissue H_2_O_2_ concentration, assessed using Amplex UltraRed assay. * *p* < 0.05 compared with non-malignant within same GS. † *p* < 0.05 compared with non-malignant in GS 3 + X group. (**B**) Immunohistochemistry representative images of anti 4-HNE on non-malignant and malignant GS 4 + X specimens. The darker brown color indicates the higher 4-HNE level. Scale bar = 50 µm.

**Figure 6 ijms-23-12168-f006:**
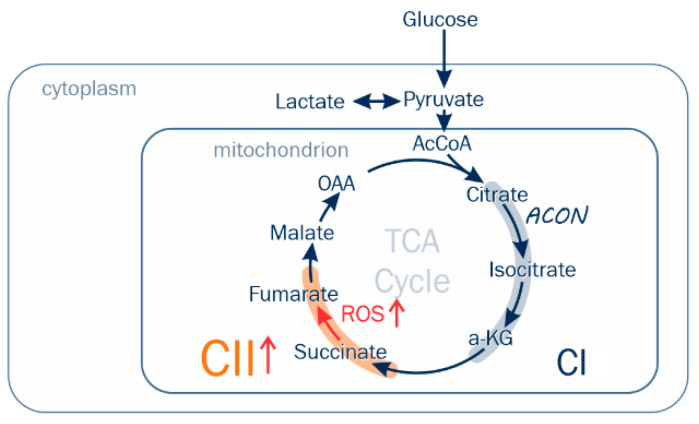
Schematic diagram of metabolic characteristics in fresh high-grade PCa tissue. P_CII_ and ROS from CII were enhanced. ACON is m-aconitase—high level zinc inhibits this enzyme in general prostate metabolism and subsequently the accumulation and secretion of citrate. α kg is α-ketoglutarate, and OAA is oxaloacetate.

**Table 1 ijms-23-12168-t001:** The respiratory results of non-malignant and malignant prostate biopsies.

Respiratory State (pmol/(s·mg))	Biopsy Paired Source			
	GS 3 + X (n = 9)		GS 4 + X (n = 6)	
	Non-Malignant	Malignant	Non-Malignant	Malignant
LEAK (L)				
CI (L_CI_)	2.14 ± 0.32	2.02 ± 0.52	2.09 ± 0.34	2.16 ± 0.98
CII (L_CII_)	4.61 ± 0.79	5.88 ± 1.43	5.27 ± 0.85	8.70 ± 2.58 *
CI + CII (L_CI+CII_)				
OXPHOS (P)				
CI (P_CI_)	5.49 ± 0.82 (33.19%) ^1^	5.14 ± 0.90 (24.68%)	5.06 ± 0.62 (32.29%)	5.34 ± 0.75 (25.05%)
CII (P_CII_)	11.26 ± 1.31 (68.08%)	12.43 ± 1.31 (59.67%)	10.79 ± 1.74 (68.86%)	13.48 ± 2.85 (63.23%)
CI + CII (P_CI+CII_)	16.54 ± 1.31	20.83 ± 3.79	15.67 ± 1.61	21.32 ± 2.25 *
ETS (E)				
CI (E_CI_)				
CII (E_CII_)	11.98 ± 1.15	13.21 ± 1.43	11.86 ± 1.38	17.44 ± 3.14
CI + CII (_ECI+CII_)	19.07 ± 1.40	23.72 ± 4.19	17.81 ± 1.38	23.98 ± 2.31 *

* *p* < 0.05 compared with paired non-malignant group; ^1^ percentage of the OXPHOS of CI + CII.

**Table 2 ijms-23-12168-t002:** Protocols for convergent CI + CII (left) and CII (right) OXPHOS assessment.

CI + CII Protocol			CII + Rotenone Protocol		
Injection	Abbreviation	Concentration	Injection	Abbreviation	Concentration
glutamate	G	10 mM	rotenone	R	0.5 µM
malate	M	2 mM	succinate	S	10 mM
ADP	D	2.5 mM	ADP	D	2.5 mM
cytochrome c	Cyt C	10 µM	cytochrome c	Cyt C	10 µM
pyruvate	P	5 mM	FCCP	F	1.5 µM
succinate	S	10 mM	malonate	Mna	5 mM
FCCP	F	1.5 µM	Antimycin A	AA	2.5 µM
rotenone	R	0.5 µM			
malonate	Mna	5 mM			
Antimycin A	AA	2.5 µM			

## Data Availability

Not applicable.

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
