# Peer review of "Enhanced Succinate Oxidation with Mitochondrial Complex II Reactive Oxygen Species Generation in Human Prostate Cancer"

_ijms, 2022, doi:10.3390/ijms232012168_

Round 1
Reviewer 1 Report
The manuscript by Zhang et al., deals with a very interesting and important issue of the regulation of prostate cancer metabolism and ATP synthesis. Moreover, the experiments are performed on fresh tissues which certainly makes the measurements even more valuable.
I only have a few comments concerning the present manuscript, in particular
1. Cancer cells, especially aggressive cancer cells are known to use Warburg effect to produce energy. Although I agree that metastatic cancers also rely on OXPHOS I think it would be good to at least discuss this issue if no data about glycolysis on the studied tissues are available. Do your data suggest potential changes of the OXPHOS/Warburg effect ratio between the tissues of different malignancy? Moreover, metabolomic analysis was performed on the tissues which means it is not possible to distinguish between mitochondria versus cytoplasm. High citrate levels are known to inhibit PFK1 and in consequences reduce glycolysis which could suggest an increased OXPHOS component. Could that be the case in some malignant tissues?
2. The GS4+X malignant tissues have relative low citrate and aconitate levels but increased the downstream intermediates compared to non-malignant tissues. Would it be possible to discuss this issue, how is it possible and where the metabolites could be synthesised?
3. Moreover, how would you explain increased citrate in low grade malignant tissues compared to other conditions, would it suggest that c-aconitase is somehow still inhibited? As you mention in the discussion, at that point blood supply should be still relatively good.
4. Any idea, why LEAK is increased in malignant tissues would that suggest decreased coupling efficiency?
Author Response
Dear editor and reviewer,
Thank you for giving us the opportunity to submit a revised draft of the manuscript “Enhanced succinate oxidation with mitochondrial complex II reactive oxygen species generation in human prostate cancer” for publication in IJMS. We appreciate the time and effort that you dedicated to providing feedback on our manuscript and are grateful for the insightful comments and valuable improvements to our paper. In this document, we address the review comments as best as possible. The edits are highlighted in red within the manuscript. Attached below are detailed responses to the reviewer’s comments. The reviewer comments are shown in black and our responses with highlighted edits and line number in red. All page numbers refer to the revised manuscript file with tracked changes.
Please let us know if you still have any questions or concerns about the manuscript. We will be happy to address them in a timely manner.
Sincerely,
The authors of ijms-1945829
Response to Reviewer 1 Comments
The manuscript by Zhang et al., deals with a very interesting and important issue of the regulation of prostate cancer metabolism and ATP synthesis. Moreover, the experiments are performed on fresh tissues which certainly makes the measurements even more valuable.
I only have a few comments concerning the present manuscript, in particular
Author response: We would like to thank you for your careful reading, helpful comments, and constructive suggestions. The edits based on your comments and points have significantly added insights into our manuscript.
Point-by-point Response to Reviewer 1 Comments
Point 1: Cancer cells, especially aggressive cancer cells are known to use Warburg effect to produce energy. Although I agree that metastatic cancers also rely on OXPHOS I think it would be good to at least discuss this issue if no data about glycolysis on the studied tissues are available. Do your data suggest potential changes of the OXPHOS/Warburg effect ratio between the tissues of different malignancy? Moreover, metabolomic analysis was performed on the tissues which means it is not possible to distinguish between mitochondria versus cytoplasm. High citrate levels are known to inhibit PFK1 and in consequences reduce glycolysis which could suggest an increased OXPHOS component. Could that be the case in some malignant tissues?
Response 1: As noted by Reviewer 1 aggressive cancers undergo a glycolytic (Warburg) switch without completely bypassing mitochondrial OXPHOS, a transition that underlies the basis for diagnostic FDG-PET imaging. Early and low-grade prostate cancers have low FDG-PET imaging sensitivity consistent with low glycolytic activity. This has also been documented in other cancers.1 Additionally the role of succinate in malignancies has also been published.2
Manuscript edits: page 8, lines 281-284.
As note by Reviewer 1, these studies were performed on permeablized tissue which limits ability for subcellular localization of metabolites. Methods for isolating mitochondrial from the fresh human prostate tissue, which is structurally complex, require larger tissue samples. Hence, to avoid disrupting the clinical pathologic interpretation on these freshly procured human samples, we mechanically permeabilized the tissue which, as noted above, limits the ability to localize metabolic analysis to cytoplasm or specific organelles such as mitochondria.
Manuscript edits: page 7, lines 240-244.
In normal prostate epithelial tissue citrate concentration and secretion are significantly elevated due to zinc inhibition of c-aconitase. Citrate is also relatively increased in lower Gleason grade PCa as documented in Figure 4. As noted by the Reviewer 1 citrate elevation suppresses PFK1 activity and reduces glycolysis which would trigger increased mitoATP synthesis. As documented in Fig 2 and Table 1 there appears to be a OXPHOS shift to increased Complex-II succinate oxidation rather than Complex-I citrate oxidation. This has been documented in other cancers as well.2,3
Manuscript edits: Page 7, line 219; Page 8, lines 286-288.
Point 2: The GS4+X malignant tissues have relative low citrate and aconitate levels but increased the downstream intermediates compared to non-malignant tissues. Would it be possible to discuss this issue, how is it possible and where the metabolites could be synthesised?
Response 2: You raise a very interesting point, one that we are giving considerable thought to. As noted above, there is a shift to complex II flux that would increase downstream fumarate, malate and OAA. This then raises the question about the source succinate if it is not from the citrate span of TCA cycle. As referenced above, this would apply to other cancers as well. We have a hypothesis that we are beginning to investigate in human PCa tissue.
Manuscript edits: Page 7, lines 240-244.
Point 3: Moreover, how would you explain increased citrate in low grade malignant tissues compared to other conditions, would it suggest that c-aconitase is somehow still inhibited? As you mention in the discussion, at that point blood supply should be still relatively good.
Response 3: Although the mechanism in not yet clear, the increased citrate concentration in lower Gleason grade PCa tissue reflects an early metabolic transformation from normal prostate epithelial tissue to a tumor microenvironment. The relative increase citrate in lower grade tissues could reflect decreased c-aconitase activity, but it could also reflect an early transformation state. For this study we did not assess enzymatic activity or isotopic tracing, hence did not discuss possibilities in this paper.
In later stages of PCa there is enhanced citrate lyase conversion of cytosolic citrate to acetyl-CoA for lipid synthesis.4,5 Hence lower tissue citrate concentration.
Manuscript edits: Page 8, lines 286-288.
Point 4: Any idea, why LEAK is increased in malignant tissues would that suggest decreased coupling efficiency?
Response 4: Insightful comment regarding the relationship of LEAK to phosphorylation coupling efficiency. The LEAK respiration results from several compensating factors that includes uncoupled proton leak into the mitochondrial matrix with reduction of protonmotive force (ΔP) as a major factor. Other factors include electron leak with superoxide (ROS) production and, to a lesser extent, electron and proton slip with diminished translocation of protons across the inner membrane. These factors result in increased compensatory oxygen consumption rate and uncoupled phosphorylation.6,7 In cancers it might reflect inner membrane permeability changes related to uncoupling proteins or altered membrane phospholipid composition.8
In this study we assessed LEAK state of respiration in the presence of CI glutamate-malate and separately with CII rotenone + succinate substrate before titrating ADP as shown in Figure 2. In consideration of the documented increase in ROS shown in Figure 5, it appears that non-phosphorylating electron leak forming superoxide and H2O2 is a major contributor to LEAK respiration in PCa. This, like proton leak, would result in decreased ΔP with increased compensatory oxygen flux but decreased coupling efficiency.
Manuscript edits: Page 3, Figure 2A corrective shift of the LCI label to the Glutamate Malate column that precedes ADP titration. There are no resultant data analysis changes.
Page 8, lines 264 – 267 clarified increased LEAK discussion as outlined above.
References:
- Long NM, Smith CS. Causes and imaging features of false positives and false negatives on F-PET/CT in oncologic imaging. Insights Imaging 2011;2:679-98.
- Dalla Pozza E, Dando I, Pacchiana R, et al. Regulation of succinate dehydrogenase and role of succinate in cancer. Seminars in cell & developmental biology 2020;98:4-14.
- Schopf B, Weissensteiner H, Schafer G, et al. OXPHOS remodeling in high-grade prostate cancer involves mtDNA mutations and increased succinate oxidation. Nat Commun 2020;11:1487.
- Zaidi N, Swinnen JV, Smans K. ATP-citrate lyase: a key player in cancer metabolism. Cancer research 2012;72:3709-14.
- Xin M, Qiao Z, Li J, et al. miR-22 inhibits tumor growth and metastasis by targeting ATP citrate lyase: evidence in osteosarcoma, prostate cancer, cervical cancer and lung cancer. Oncotarget 2016;7:44252-65.
- Brand MD, Chien LF, Ainscow EK, Rolfe DF, Porter RK. The causes and functions of mitochondrial proton leak. Biochimica et biophysica acta 1994;1187:132-9.
- Cheng J, Nanayakkara G, Shao Y, et al. Mitochondrial Proton Leak Plays a Critical Role in Pathogenesis of Cardiovascular Diseases. Advances in experimental medicine and biology 2017;982:359-70.
- Baffy G, Derdak Z, Robson SC. Mitochondrial recoupling: a novel therapeutic strategy for cancer? British journal of cancer 2011;105:469-74.
Reviewer 2 Report
Prostate cancer is the second most commonly diagnosed cancer and the fifth leading cause of cancer death in men worldwide. Well-regulated and highly specialized citrate-orientated metabolism is maintained in prostatic epithelial cells, while recent studies showed a transition from citrate secretion to citrate oxidation caused by zinc abnormal accumulation happens when prostate cancer (PC) occurs. Moreover, further studies revealed mitochondrial homeostasis impact PC tumorigenesis by regulating β-oxidation, ROS level and electron transport activity. However, previous studies mainly relied on pre-clinical models, established cell lines, and fixed or frozen tissue bank samples, the mechanical properties of PC metabolism still need further exploration in fresh PC tissue.
In this study, Aijun Zhang et al. compared mitochondrial respiratory function in paired non-malignant and PC specimens from freshly procured human prostate, and found mitochondrial CII succinate oxidation and ROS generation are boosted in fresh PC tissue. Overall, the findings in this research are potentially interesting, and the paper is generally well written, but the underlying mechanisms are not clear and more evidence is necessary to substantiate the authors' conclusions.
I have two major concerns. First, the underlying mechanisms, both in metabolism and transcriptional control, contribute to the enhanced PCII and impaired PCI during to PC malignant transformation is unclear. Second, Bernd Schöpf et al (PMCID: PMC7083862). recently, found “high levels of potentially deleterious mutations in mitochondrial Complex I-encoding genes are associated with a 70% reduction in NADH-pathway capacity and compensation by increased succinate-pathway capacity”, and “upregulation of CII as an important mechanism to enable sustained OXPHOS capacity for ATP production in tumors despite CI protein mutations affecting CI and N-pathway contribution”. Here, authors found similar results although fresh PC specimens were used here. Novel in-depth mechanisms involved in metabolic transform in PC tumorigenesis need to be uncovered with fresh samples.
Moreover, several concerns need to be addressed to improve the quality of the manuscript.
1) Line 71-73: “Each biopsy was split into three sections: one for high-resolution respirometry (HRR) study, one for mRNA expression, and a third for histopathological diagnostics.” So what’s the difference in mRNA expression between biopsies?
2) It would be better to put a simple model in Fig.2 to show the CI and CII pathways.
3) Table 1 should be moved to supplemental data.
4) The oxygen concentration/consumption curves of non-malignant/malignant can be merged into one panel in Figure 2A(B), in order to compare the difference between the two samples more directly.
Round 2
Reviewer 2 Report
I am satisfied with the authors' modifications after the first-round review. However, the manuscript needs to be double-checked to avoid careless mistakes, e.g. : "2.3 Histopathology and immunohistochemistry" (line 361) should be "4.3 Histopathology and immunohistochemistry".
Author Response
We appreciate the Reviewer 2 detailed and timely response. We have edited the downloaded revision manuscript and will re-submit it.
Thank you